# Migration-Related Weight Changes among African Immigrants in the United States

**DOI:** 10.3390/ijerph192315501

**Published:** 2022-11-23

**Authors:** Samuel Byiringiro, Binu Koirala, Tiwaloluwa Ajibewa, Eric K. Broni, Xiaoyue Liu, Khadijat Adeleye, Ruth-Alma N. Turkson-Ocran, Diana Baptiste, Oluwabunmi Ogungbe, Cheryl Dennison Himmelfarb, Serina Gbaba, Yvonne Commodore-Mensah

**Affiliations:** 1School of Nursing, Johns Hopkins University, Baltimore, MD 21205, USA; 2Feinberg School of Medicine, Northwestern University, Chicago, IL 60611, USA; 3Johns Hopkins Ciccarone Center for the Prevention of Cardiovascular Disease, Baltimore, MD 21093, USA; 4Elaine Marieb College of Nursing, University of Massachusetts, Amherst, MA 01003, USA; 5Division of General Medicine, Beth Israel Deaconess Medical Center, Boston, MA 02215, USA

**Keywords:** body mass index, African immigrants, body weight

## Abstract

(1) Background: people who migrate from low-to high-income countries are at an increased risk of weight gain, and excess weight is a risk factor for cardiovascular disease. Few studies have quantified the changes in body mass index (BMI) pre- and post-migration among African immigrants. We assessed changes in BMI pre- and post-migration from Africa to the United States (US) and its associated risk factors. (2) Methods: we performed a cross-sectional analysis of the African Immigrant Health Study, which included African immigrants in the Baltimore-Washington District of the Columbia metropolitan area. BMI category change was the outcome of interest, categorized as healthy BMI change or maintenance, unhealthy BMI maintenance, and unhealthy BMI change. We explored the following potential factors of BMI change: sex, age at migration, percentage of life in the US, perceived stress, and reasons for migration. We performed multinomial logistic regression adjusting for employment, education, income, and marital status. (3) Results: we included 300 participants with a mean (±SD) current age of 47 (±11.4) years, and 56% were female. Overall, 14% of the participants had a healthy BMI change or maintenance, 22% had an unhealthy BMI maintenance, and 64% had an unhealthy BMI change. Each year of age at immigration was associated with a 7% higher relative risk of maintaining an unhealthy BMI (relative risk ratio [RRR]: 1.07; 95% CI 1.01, 1.14), and compared to men, females had two times the relative risk of unhealthy BMI maintenance (RRR: 2.67; 95% CI 1.02, 7.02). Spending 25% or more of life in the US was associated with a 3-fold higher risk of unhealthy BMI change (RRR: 2.78; 95% CI 1.1, 6.97). (4) Conclusions: the age at immigration, the reason for migration, and length of residence in the US could inform health promotion interventions that are targeted at preventing unhealthy weight gain among African immigrants.

## 1. Introduction

Obesity is a major global health challenge. Over 650 million adults were living with obesity in 2016 [1]. In the same year, obesity-related aggregated medical costs among adults in the United States (US) were $260.6 billion [1]. Obesity is strongly linked with all forms of cardiovascular disease (CVD) and risk factors, such as hypertension, diabetes, and dyslipidemia, among other chronic diseases [2]. Black individuals in the US are disproportionately affected by obesity [3].

A potent relationship exists between obesity and urbanization/industrialization [4,5]. A cross-sectional study on obesity and diabetes conducted among middle-aged Ghanaians living in rural and urban Ghana and three European cities found a “gradient of rising prevalence of obesity from rural through urban Ghana to Europe” [4]. In this study, the prevalence of obesity was up to 15 times higher among Ghanaian migrants in Europe compared to their rural counterparts in Ghana. A systematic review by Commodore-Mensah et al. reported an overweight/obesity prevalence range from 20 to 62% and 4 to 49% among Ghanaians and Nigerians resident in their home countries, respectively [5]. However, an overweight/obesity prevalence of 66–90% was reported among Ghanaian and Nigerian immigrants in industrialized countries [5].

Between 2010 and 2018, the sub-Saharan African immigrant population in the US increased by 52%, which was over four times the 12% growth rate for the US immigrant population during that period [6]. Immigrants from Nigeria, Ghana, Sierra Leone, and Liberia alone constitute more than one-third of US African immigrants [7]. Between 2013 and 2017, the US Census Bureau reported 181,000 African immigrants in the Washington, District of the Columbia (DC) metropolitan area. The African immigrants represented 13% of the area’s total immigrant population, which was three times the national average of 4.5% [7].

The factors that influence body weight include non-modifiable ones, such as the person’s genetic makeup, age, and sex [8]. Modifiable factors of body weight are physical activity, diet, and some environmental and social factors, such as access to safe places for physical activity and body image perceptions in society [8]. Migrants undergo alterations in their physical and social environments, which affects their access to food, resources for physical exercise, and social networks; this causes stress from the transition and adjustment into the new environment. These changes could have a big effect on migrants’ body weight and overall health. The literature suggests that when exposed to an obesogenic environment, migrants are more vulnerable to weight gain than native residents [9].

Acculturation, which is the process of adapting and acquiring the traits of a new culture, has been proposed as a major explanation for weight gain when moving to an obesogenic environment [9]. According to the acculturation concept, older people are likely to cling to their original culture and find difficulties adjusting to the new culture, while younger individuals are likely to adopt the new culture [10]. Additionally, longer exposure to a new culture often results in a higher likelihood of adopting many aspects of that culture [10]. In reference to the acculturation process, people who migrate at a young or middle age would have a high risk of weight gain, while migration at an older age would likely result in the maintenance of pre-migration weight.

Despite the difference in overweight/obesity prevalence among Africans on the continent and those in industrialized countries, the literature is scarce on pre- and post-migration related weight changes in African immigrants in the US. Thus, we investigated migration-related weight changes and associated factors among African immigrants living in the Washington DC metropolitan area.

## 2. Materials and Methods


**Participants**


The African Immigrant Health Study (AIHS) included African immigrants dwelling in the Washington DC metropolitan area and was conducted between 2017 and 2019. Participants completed a one-time self-administered survey and had their anthropometric measurements taken by research assistants. Potential participants were recruited from religious places of worship (churches and mosques) and community-based organizations [11]. Eligible participants were 30 years and older at the time of data collection, not pregnant, able to communicate in English, and first-generation immigrants to the US from Cameroon, Liberia, Ghana, Nigeria, or Sierra Leone. An institutional review board approved this study (IRB00129394, Approved 23 February 2017). A detailed description of the methods and data collection procedure was published elsewhere [11].

A total sample of 465 participants was recruited and joined the study. One hundred and fourteen participants were excluded from the analysis because they were missing weight premigration or age at migration. An additional 51 participants were excluded from the analysis for being in the underweight category before or after migration. The remaining 300 participants were included in the current analysis.


**Design**


We used a cross-sectional design to examine the weight change and associated factors among African Immigrants to the US.


**Outcome of interest**


The outcome of interest in this study was the change in the body mass index (BMI) category pre- and post-migration. We assessed pre-migration BMI using the participants’ self-reported pre-migration weight and current height. Participants 18 years or below at the time of immigration were excluded from the analysis because they were likely to be still growing, and their pre-migration height could distort our estimation of BMI pre-migration using their current height. We calculated the post-migration BMI from the measured weight and height. BMI categories pre- and post-migration used the Quetelet’s index: underweight (<18.5 kg/m^2^), normal weight (18.5–24.9 kg/m^2^), overweight (25.0–29.9 kg/m^2^), and obesity (≥30.0 kg/m^2^) [12]. We transformed pre- and post-migration BMI change into a categorical variable: (i) Healthy BMI change or maintenance defined as maintaining a normal BMI and changing from overweight/obesity to normal, or from obesity to overweight; (ii) Unhealthy BMI maintenance—maintaining overweight or obese BMI categories; and (iii) Unhealthy BMI change—changing from normal to overweight/obesity or from overweight to obesity BMI categories. Participants categorized as underweight before or after migration were excluded from the current analysis because an underweight status is predominantly associated with underlying health conditions [13,14,15]. We also excluded people with a missing weight, height, or age at migration.


**Exposures**


The exposures of interest were age at migration, sex, percentage of life spent in the US, stress level, and reasons for migration. The age at migration was estimated by subtracting the participants’ year of immigration to the US from their birth year. The age at migration was analyzed as a continuous variable. Sex was self-reported and categorized as male or female. The percentage of life in the US was calculated by dividing the number of years lived in the US by the current age of the participants. To be consistent with the prior literature, which reports that weight gain tends to increase significantly after 10 years of migration [16,17,18,19], we calculated the corresponding average percentage of life in the US and found 25%. We hence dichotomized the percentage of life in the US into ≥25% or <25%. 

Additionally, we assessed the association between self-reported stress and BMI changes. We used the four-item perceived stress scale (PSS-4) to measure the participant’s stress levels [20]. PSS-4 scores ranged from 0 to 16, and higher scores indicated a high level of stress. PSS-4 scores were dichotomized using the population mean score of six, where people with a score of six or higher were classified as having high levels of stress and less than six as having low levels of stress [20]. We additionally explored the association between weight change and the participants’ reasons. We categorized the reasons for migration as follows: education, family, employment/economy, asylum-seeking or refugee status, and other reasons.


**Covariates**


The self-reported covariates explored were marital status, employment status, educational level, and income level. Marital status was categorized as never married, married, or cohabiting, and separated, widowed, or divorced. The employment status was recorded and dichotomized as employed or not employed. The educational level was categorized as high school or less, college, bachelor’s degree, and graduate. The annual household income level was categorized as less than $40,000; $40,000–$70,000; $70,000–< 100,000; and ≥$100,000.


**Statistical Analyses**


We examined the sociodemographic characteristics by the BMI change category and used means (±SD) to summarize the continuous variables, proportions, and percentages for the categorical variables. We used a one-way analysis of variance with the Bonferroni correction of the alpha error level to assess the differences in the means of continuous variables across the three BMI category change directions. We used the chi-square test to compare proportions. The change in the proportions of people in the different BMI categories before and after the migration was reported using a Sankey plot.

We used multinomial logistic regression models to examine the association between the biological exposures (age at migration and sex), acculturation exposures (percentage of life in the US), stress level, and the reason for migration with the BMI change category. The bivariate regression analysis between each exposure and BMI change category was followed by the full regression model adjusting for marital status, employment status, educational level, and income level. Analyses were performed using the Stata/IC 16.1 version, and associations with *p*-values of 0.05 or less were considered statistically significant. 

## 3. Results

### 3.1. Sociodemographic Characteristics

Among 300 participants, the mean current age (±SD) was 47 (±11.4) years, and the mean (SD) age at migration was 33 (10.6) (Table 1). There were significant differences in the current age and age at immigration across the different BMI change categories, where people in the Unhealthy BMI maintenance category tended to be of older age. Female participants constituted 56% of the sample, almost three-quarters (74%) of all the participants were married or cohabiting, and 195 (65%) had a bachelor’s degree or higher level of education. The main three reasons for migration were family, 87 (30.5%), education, 74 (26%), and employment or economic hardships, 70 (24.5%). Most participants were from Ghana, 117 (39%), and Nigeria, 99 (33%). One hundred and seventy (55.6%) participants had spent 25% or more of their lives in the US. There were significant differences in employment status, percentage of life in the US, and reason for migration across the three categories of BMI change.

### 3.2. BMI, BMI Category, and Weight Changes Post Migration

Prior to migration, 54.7% had a normal BMI, 32.3% had a BMI in the overweight category, and 13.0% had obesity (Figure 1). Post-migration, 54% of the respondents had obesity, while 12% had a normal BMI. The proportion of participants in the overweight category remained roughly one-third before and after migration. People who were young at the time of migration demonstrated a wider change in BMI post-migration (Figure 2). Regarding the BMI category change post-migration, 14% had a healthy BMI change or maintenance, 22% had an unhealthy BMI maintenance, and 63.7% of the participants had an unhealthy BMI change. In terms of weight change, 48% of the participants gained more than 30 lbs after migrating to the US.

### 3.3. Factors Associated with BMI Category Changes

At the time of migration, each one-year increase in age was associated with a 7% higher relative risk of unhealthy BMI maintenance (relative risk ratio [RRR]: 1.07; 95% CI 1.01, 1.14) after adjusting for all other covariates (Table 2). Additionally, compared to men, women had two times the relative risk of unhealthy BMI maintenance (RRR: 2.67; 95% CI 1.02, 7.02).

Individuals with perceived stress scores above six had a two-fold relative risk of an unhealthy BMI change (RRR: 2.24; 95% CI 1.01, 4.99) and, compared to immigration for employment or economic reasons, immigration for educational reasons was associated with four times higher relative risk of unhealthy BMI change (RRR: 3.88; 95% CI 1.18, 12.75). Additionally, immigrants who had stayed in the US for 25% or more of their lives had almost three times the relative risk of an unhealthy BMI change than their counterparts who had spent less than a quarter of their lives in the US (RRR: 2.78; 95% CI 1.1, 6.97).

## 4. Discussion

This study explored the migration-related weight changes among African immigrants living in the Baltimore-Washington DC, metropolitan area. We found that 48% of immigrants had gained more than 30 lbs.; hence, 88% were overweight/obese after moving to the US. Both older age at migration and the female sex were associated with unhealthy BMI maintenance and higher perceived stress, migration for educational reasons, and a longer stay in the US were associated with unhealthy BMI change.

Migration to the US was associated with a BMI increase among 64% of the immigrants. The current findings are in line with prior studies that have demonstrated that migration from African nations to the US and other high-income countries [21], as well as the urbanization of African cities, are associated with weight gain. Gona and colleagues found that among Zimbabwean immigrants in the US, three-quarters were overweight or obese [22]. Similarly, Njeru and colleagues found high rates of overweight and obesity in a large sample of Somali migrants and refugees in the US, wherein 41% of male immigrants, 64% of female immigrants, and 50% of their total sample were classified as overweight or obese [23]. The Research on Obesity and Diabetes among African Migrants (RODAM) reported a 23% prevalence of overweight/obesity among the rural residents of Ghana compared to 59.8% among the urban residents in Ghana and 83% among Ghanaian immigrants to London, United Kingdom [24]. A recent systematic review on the cardiometabolic health of African immigrants to high-income countries reported a 59% (95% CI: 44–73%) overall prevalence of overweight/obesity among African immigrants to high-income countries [21], further highlighting the role of immigration on weight change among African immigrants.

We found that migration for educational reasons was associated with an unhealthy BMI change. These findings have been supported by Deforche and colleagues’ prospective study, which among other things, explored weight changes during the transition to higher education [25]. In one and a half years of follow-up, students gained an average of 2.7 kg [25]. Similar findings of an increased BMI among college students have been reported by Deng and colleagues [26]. Still, many studies report that higher education is associated with a lower BMI [27,28]. The lower BMI is attributed to a higher social-economic status often associated with higher education [29]. Although higher socioeconomic status is continuously associated with lower BMI in developed countries, studies in low-income countries show the opposite findings, where wealthy people tend to have higher BMI. The effect of education on BMI change among African immigrants warrants further examination.

Older age at the time of migration and being a woman were associated with unhealthy BMI maintenance. These findings could be partially explained by acculturation. As immigrants immerse themselves into the host culture, there is a gradual exchange between the immigrant’s original values, attitudes, and behaviors and those of the host country [30]. The process of acculturation could result in poor mental well-being and the adoption of unhealthy behavior, which consequently increases the risk of becoming overweight or obese [30,31]. Although immigrants are found to exhibit better health outcomes than domestic-born populations, a large body of evidence has shown a decline in their health over time [32]. Our findings on the relationship between age at migration and unhealthy BMI maintenance are particularly important, as middle-aged immigrants may be exposed to cumulative levels of acculturative stress that impact their weight management. In addition, the literature supports our results on the female sex as a determinant of unhealthy BMI maintenance [33]. Among North African immigrants in Europe, researchers found a higher prevalence of overweight and obesity among females than males [33]. Another study reported that adopting poor dietary habits, acculturative pressure, lack of nutritional information, and lack of family support were major barriers to weight loss among Mexican-immigrant women [34].

Our findings highlight that spending 25% or higher of life in the US is associated with unhealthy BMI change. The US has a uniquely obesogenic environment, which can contribute to unhealthy weight gain if exposed to it for a prolonged period of time. The prevalence of obesity among individuals aged 15 years and older in the US is 38%, higher than any other industrialized country [35] and far higher than the African immigrants’ countries of origin. Multiple factors could help explain the obesogenic US environment, including food insecurity among high-risk populations, including immigrants [36], the ubiquity and high marketing of multiple fast-food franchises [37], reliance on driving as the only means of transport [38], lack of access to safe places for physical exercise and so on [38]. Yet, in most countries of Africa, weight gain factors, such as driving a car, eating fast food, and the notion of gaining weight, have been predominantly regarded as the signs of wealth and being healthy [39,40]. The notion of regarding the obesogenic factors as signs of wealth could help explain why immigrants (from low-income settings) could be more negatively affected by the obesogenic environment than the local dwellers of that environment. More studies are needed to elicit the additional underlying factors that prevent African immigrants from maintaining a healthy BMI. More importantly, it is essential to identify the interventions tailored to high-risk populations to improve their weight outcomes.

Alas, the environment is not the only factor for unhealthy weight management among immigrants. In our findings, perceived stress was associated with unhealthy BMI change. Prior evidence from a double-blinded randomized controlled trial has demonstrated that high stress leads to low post-meal energy expenditure, which favors the development of overweight and obesity [41]. Migrants are exposed to different stressors, including financial stressors, the fear of deportation, acculturation stress, being far from their family and loved ones, language barriers, and so on [42,43,44]. An additional challenge for African immigrants living in the US is that they have lower utilization of mental health services despite a greater need for those services [44,45]. These stressors could not only interfere with the African immigrants’ ability to maintain healthy weight management practices but also alter their food metabolism, leading to weight gain. Interventions to address African immigrants’ mental health could have a positive effect on their cardiometabolic health outcomes.

There are limitations to the current study. The cross-sectional nature of this study limits our ability to explore the temporal effect of migration on weight changes before and after migration. Additional weight measurements on the same individuals two to three years after the first measurement would provide stronger evidence regarding the association between weight change and the length of stay in the US. Additionally, we based the weight change on the objectively measured weight post-migration and the self-reported weight prior to migration which could introduce the recall bias. The risk for recall bias increases among people who have lived in the US for more extended periods of time. Potential strategies to deal with the recall bias could be to recruit participants in their first year of arrival in the US and follow them up prospectively. Furthermore, strong evidence exists on the effect of physical activities and nutrition behaviors on weight change [46], yet data on these two variables were self-reported or had predominantly missing responses, and we could not include those variables in the current analysis. Finally, BMI is an imperfect measure of adiposity because it does not capture visceral adiposity [47] and is derived from a Eurocentric population. Our study, however, was more concerned with exploring the trend of weight change pre- and post-migration. Despite the limitations, this study is, to our knowledge, one of the first to explore weight changes before and after migration from Africa to the US and its associated factors. This study will contribute to the literature and health equity efforts by informing targeted policies and campaigns to mitigate overweight/obesity and its CVD- related risk in this rapidly growing population in the US.

## 5. Conclusions

In conclusion, migration from Africa to the US is associated with extensive weight gain, especially among females who migrate for educational reasons or who are in their middle adulthood. Future studies should prospectively validate the risk populations (high-risk populations for weight gain) and implement interventions to address this epidemic.

## Figures and Tables

**Figure 1 ijerph-19-15501-f001:**
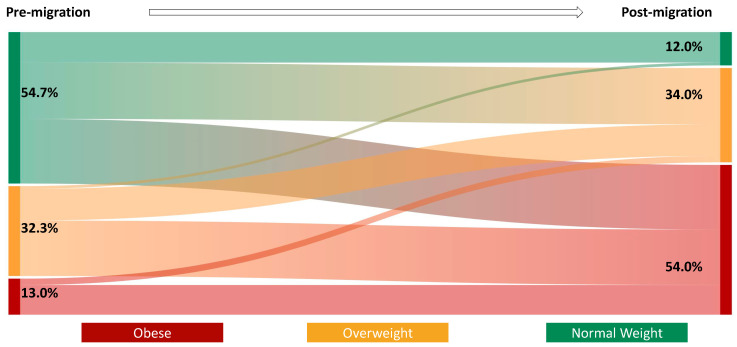
Body mass index change pre- and post-migration to the US.

**Figure 2 ijerph-19-15501-f002:**
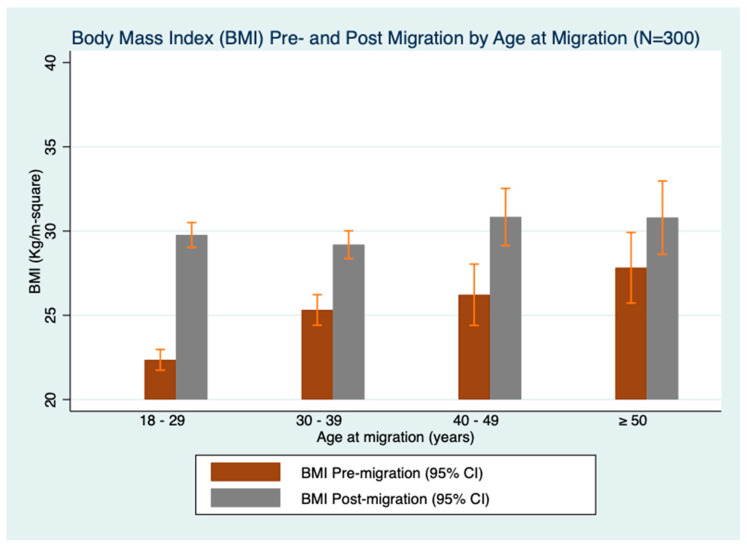
Body Mass Index pre- and post-migration to the US by age at migration.

**Table 1 ijerph-19-15501-t001:** Demographic characteristic of participants (N = 300).

Variable	Total (N = 300)	Healthy BMI Change or Maintenance N = 43	Unhealthy BMI Maintenance N = 66	Unhealthy BMI Change N = 191	*p*-Value
Current age, mean (SD)	46.89 (11.4)	42.57 (10.7) ^a^ *	49.44 (11.2) ^b^	46.97 (11.5) ^b^	0.009 ^†^
Age at migration, mean (SD)	33.03 (10.6)	33.30 (9.4) ^a^ *	40.39 (11.0) ^b^	30.42 (9.41) ^ac^	0.000 ^†^
Sex					0.119
Male	132 (44.00)	25 (58.14)	26 (39.39)	81 (42.41)	
Female	168 (56.00)	18 (41.86)	40 (60.61)	110 (57.59)	
Marital status					0.833
Never married	26 (8.67)	5 (11.63)	4 (6.06)	17 (8.9)	
Married or cohabiting	222 (74.00)	32 (74.42)	49 (74.24)	141 (73.82)	
Separated/widowed/divorced	53 (17.33)	6 (13.95)	13 (19.7)	33 (17.28)	
Educational level					0.447
Less than bachelor’s degree	104 (34.78)	14 (32.56)	21 (31.82)	69 (36.32)	
Bachelor’s degree	102 (34.11)	18 (41.86)	19 (28.79)	65 (34.21)	
Graduate	93 (31.10)	11 (25.58)	26 (39.39)	56 (29.47)	
Income level					0.360
<$40,000	52 (18.51)	12 (28.57)	12 (20.69)	28 (15.47)	
$40,000–<$70,000	72 (25.62)	8 (19.05)	15 (25.86)	49 (27.07)	
$70,000–<$100,000	68 (24.20)	9 (21.43)	10 (17.24)	49 (27.07)	
≥$100,000	89 (31.67)	13 (30.95)	21 (36.21)	55 (30.39)	
Currently employed					0.002 ^†^
Yes	246 (83.39)	7 (16.28)	20 (30.77)	22 (11.76)	
No	49 (16.61)	36 (83.72)	45 (69.23)	165 (88.24)	
Percentage of life in the US					0.000 ^†^
<25%	129 (43.00)	27 (62.79)	44 (66.67)	58 (30.37)	
25% or more	171 (57.00)	16 (37.21)	22 (33.33)	133 (69.63)	
Reason for migration					0.034 ^†^
Education	74 (25.96)	5 (11.9)	11 (17.19)	58 (32.4)	
Employment/Economy	70 (24.56)	17 (40.48)	15 (23.44)	38 (21.23)	
Family reasons	87 (30.53)	13 (30.95)	22 (34.38)	52 (29.05)	
Asylum/Refugee	21 (7.37)	2 (4.76)	8 (12.5)	11 (6.15)	
Other reasons	33 (11.58)	5 (11.9)	8 (12.5)	20 (11.17)	

^†^*p*-value < 0.05; * Different superscripted letters represent statistical significance between group differences in means.

**Table 2 ijerph-19-15501-t002:** Unadjusted and adjusted multinomial logistic regression of factors associated with body mass index change among African immigrants to the US. N = 258.

	Model 1	Model 2
**Healthy BMI change or maintenance**	1 (ref)	1 (ref)
**Unhealthy BMI maintenance**		
Age at migration, years	1.06 (1.02, 1.10) **	1.07 (1.01, 1.14) *
Sex		
Male	1 (ref)	1 (ref)
Female	2.13 (0.98, 4.67)	2.67 (1.02, 7.02) *
Stress level		
Low (PSS < 6)	1 (ref)	1 (ref)
High (PSS ≥ 6)	2.06 (0.92, 4.64)	1.96 (0.77, 4.99)
Reason for migration		
Employment/Economy	1 (ref)	1 (ref)
Education	2.49 (0.70, 8.83)	2.55 (0.61, 10.61)
Family reasons	1.92 (0.72, 5.09)	1.11 (0.34, 3.65)
Asylum/Refugee	4.53 (0.83, 24.76)	3.06 (0.43, 21.78)
Others	1.81 (0.49, 6.76)	1.29 (0.27, 6.04)
Percentage of life in the US		
<25%	1 (ref)	1 (ref)
≥25%	0.84 (0.38, 1.88)	1.16 (0.39, 3.45)
**Unhealthy BMI change**		
Age at migration, years	0.97 (0.93, 1.00)	1.00 (0.94, 1.06)
Sex		
Male	1 (ref)	1 (ref)
Female	1.88 (0.96, 3.68)	1.75 (0.78, 3.92)
Stress level		
Low (PSS < 6)	1 (ref)	1 (ref)
High (PSS ≥ 6)	2.13 (1.05, 4.33) *	2.24 (1.01, 4.99) *
Reason for migration		
Employment/Economy	1 (ref)	1 (ref)
Education	5.19 (1.76, 15.24) **	3.88 (1.17; 12.75) *
Family reasons	1.79 (0.77, 4.12)	1.23 (0.46, 3.24)
Asylum/Refugee	2.46 (0.49, 12.32)	2.15 (0.35, 13.03)
Others	1.79 (0.57, 5.56)	1.3 (0.36, 4.85)
Percentage of life in the US		
<25%	1 (ref)	1 (ref)
≥25%	3.77 (1.89, 7.52) **	2.78 (1.1, 6.97) *

Model 1: Bivariate analysis of each variable with the BMI change category. Model 2: Included all variables of Model 1 and adjusted for employment, education, income, and marital status. ** *p* < 0.01, * *p* < 0.05. BMI category definitions: Normal: 18.5–24.9 kg/m^2^; Overweight: 25.0–29.9 kg/m^2^; Obesity: ≥30.0 kg/m^2^. Healthy BMI change or maintenance: maintaining normal BMI, changing to normal from overweight/obesity, or from obesity to overweight; Unhealthy BMI maintenance: BMI maintenance of overweight or obesity; Unhealthy BMI change: BMI category change from normal to overweight/obesity, or from overweight to obesity.

## Data Availability

The datasets generated during and/or analyzed during the current study are available from the corresponding author on reasonable request.

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
