# Peer review of "Migration-Related Weight Changes among African Immigrants in the United States"

_ijerph, 2022, doi:10.3390/ijerph192315501_

Round 1

Reviewer 1 Report

The manuscript “Migration-related weight changes among African Immigrants in the United States” by Byiringiro et al provides a cross-sectional analysis of African immigrants in the Baltimore-Washington-DC area. The authors conclude that a number of factors, including the age at and the reason for migration, as well as the duration of stay in the US could serve as relevant indicators predicting unhealthy weight gain among African immigrants. The identification of such indicators is very important in guiding general public and health care providers towards the development of tailored life-style changes and therapeutic interventions against obesity and obesity-related diseases.

The study is well designed; its limitations are properly acknowledged. To the reviewer, major limitations (which might be better to discuss at length) include its cross-sectional nature (it would be very informative to provide some sort of longitudinal analysis, for example, taking a second data point 2-3y apart from the first one – to test the conclusion regarding the length of stay in the US), and self-reported assessment of the pre-migration weight (some ways to correct for the recall bias need to be discussed and proposed for future studies). Furthermore, potential effects of the age-associated increase of the body weight need to be taken into account: for example, with the graph showing the age data and the weight changes for individual study participants, with horizontal lines showing individual age data (migration/current) with different colors indicating BMI changes.    

Minor points:

lines 139-140: please modify the sentence (is a bit unclear in its current form)

line 178: a typo (“participants”)

line 238: a typo (repeated word)

line 273: a typo (“excise”)

Reviewer 2 Report

Language and technical care:

The manuscript requires some minor attention in terms of overall language and technical aspects, with a few examples highlighted below:

-          Line 21 – extra space before the word “Background”;

-          Line 24 – add a space after the word “pre-“;

-          Some sections of the document are justified to the full page, while other sections not, see paragraph 3.2 and 3.3;

-          Line 129 - full-stop missing after the reference;

-          Line 135 – remove additional space before the word “Sex”;

-          Line 140 – should the word “we” be caps – is this the beginning of a new sentence?;

-          Line 173 – remove the additional space between the words in the heading;

-          Line 174 – remove the additional space before the word “the” after the word “participants,”;

-          Line 174 – insert space after the value 47;

-          Line 178 – spelling of participants;

-          Line 190 – why is the word Figure 1 in bold;

The manuscript is very well referenced using relevant, up-to-date references.

-          Line 60 – is the reference for this line the same as the previous?;

Literature Review:

This reviewer felt that the literature review is comprehensive and situates/contextualises all the relevant aims and objectives of the research.

Methodology and materials:

The reviewer believes that the methodology is straightforward and explained well, and good and acceptable research procedures have been followed.

-          It feels as if there is no clarity regarding the age of eligibility – in line 100 it states that participants needed to be 30 years and older, while line 116 says 18 years?

Results and Discussion:

The reviewer believes that the results and discussion of the results are all scientifically sound, and the presentation of the results are done well. The graph in line 198 is difficult to read in grey-scale colouring, and would probably be easier to interpret and understand when presented in colour. But overall an excellent presentation of results.

Conclusion:

The reviewer believes that the conclusion is presented adequately and the worthiness of the research is evident, particularly in light of our never-ending fight against the global onslaught of diet-related illnesses amongst people living in urban areas. It would be very interesting to see how similar results differ for immigrants all over the world.

Reviewer 3 Report

The authors investigate migration-related weight changes among African immigrants living in the Baltimore-Washington DC metropolitan area and identify potential risk factors. They conclude that future studies are needed to further identify risk factors which are essential to target interventions to high-risk populations to improve their weight outcomes.

The article is relevant. The paper is straightforward, well-written and well structured. Methods are appropriate, are adequately described and well-executed. The study’s limitations have been acknowledged. Finally, the results can be useful.

I have some minor comments which I list below:

·       I would suggest including the significant differences in Table 1.

·       The relationship between spending 25% or more of life in the US and higher risk of unhealthy BMI change, between individuals with perceived stress scores and unhealthy BMI change, and between immigration for educational reasons and higher relative risk of unhealthy BMI change have not been discussed in the discussion. I would suggest doing so.
